# Quasi-one-dimensional metallic conduction channels in exotic ferroelectric topological defects

Wenda Yang[1,4], Guo Tian[1,4], Yang Zhang [2,3], Fei Xue[2], Dongfeng Zheng[1], Luyong Zhang[1], Yadong Wang[1], Chao Chen[1], Zhen Fan [1], Zhipeng Hou[1], Deyang Chen [1], Jinwei Gao[1], Min Zeng[1], Minghui Qin[1], Long-Qing Chen [2], Xingsen Gao [1✉] & Jun-Ming Liu[1,3]

Ferroelectric topological objects provide a fertile ground for exploring emerging physical properties that could potentially be utilized in future nanoelectronic devices. Here, we demonstrate quasi-one-dimensional metallic high conduction channels associated with the topological cores of quadrant vortex domain and center domain (monopole-like) states confined in high quality $BiFeO_3$ nanoislands, abbreviated as the vortex core and the center core. We unveil via the phase-field simulation that the superfine metallic conduction channels along the center cores arise from the screening charge carriers confined at the core region, whereas the high conductance of vortex cores results from a field-induced twisted state. These conducting channels can be reversibly created and deleted by manipulating the two topological states via electric field, leading to an apparent electroresistance effect with an on/off ratio higher than $10^3$. These results open up the possibility of utilizing these functional one-dimensional topological objects in high-density nanoelectronic devices, e.g. nonvolatile memory.

[1] Guangdong Provincial Key Laboratory of Quantum Engineering and Quantum Materials, Institute for Advanced Materials, South China Academy of Advanced Optoelectronics, South China Normal University, Guangzhou, China. [2] Department of Materials Science and Engineering, The Pennsylvania State University, University Park, PA, USA. [3] Laboratory of Solid-State Microstructures and Innovation Center of Advanced Microstructures, Nanjing University, Nanjing, China. [4]These authors contributed equally: Wenda Yang, Guo Tian. ✉email: xingsengao@scnu.edu.cn

Topological objects and defects (e.g. domain walls, vortices, skyrmions) in condensed matters have garnered massive attention as an arena of exploring emerging exotic phenomena and functionalities[1–3]. In materials with ferroic order, these topological objects can also be manipulated and controlled by external fields without disrupting their host lattice, making them promising elemental building blocks for potential configurable topological nanoelectronics[3–6]. To this stage, most earlier investigations have hitherto focused on the properties of two-dimensional (2D) defects (namely domain walls). For instance, a plethora of physical properties have been observed in ferroelectric domain walls, including enhanced domain wall conductivities[7–10], enhanced photovoltaics[11], giant magnetoresistances[12], extraordinary magnetism[13,14], and quantum oscillation behaviors[15], among many others. These functionalities could underpin a wide range of potential configurable nanoelectronic, magnetoelectronic, and optoelectronic applications, particularly the energy-efficient current-readout nonvolatile memories based on reversible creating/eliminating of the conductive domain walls[16,17]. However, such conceptual devices usually face the obstacles of low scalability (domain wall is a 2D defect) and unstable restoration process due to the difficulty in deterministic control of the walls. Recent studies did show the enhanced repeatability of domain wall switching in center-type domains due to the topological and geometric restriction effects[18]. This motivation, in practice, could suffer from the non-uniformity in domain wall conductivity arisen from local distortions (e.g., bending or tilting)[19], detrimental to device performance.

In recent years, there has been increasing interest in more complex ferroelectric topological objects, leading to the discovery of a series of topological states, such as closure domain states[20–24], quadrant vortex states[25–28], circular vortices lattices[29], skyrmions[30], meron states[31], bubble domains[32], and center domain states (monopole-like structure with polarization pointing inward/outward the core)[18,33,34] in size-confined thin films/superlattices/nanostructures, as well as unique multifold vortex structures in improper ferroelectrics[35,36] and organic ferroelectrics[37]. These tantalizing findings have kindled the excitement for exploring device concepts associated with these exotic topological defects. For example, exotic one-dimensional (1D) topological defects, e.g., vortex or center domain cores, take the advantages of 1D superfine dimensionality and topological protection nature (resilience against perturbations), which not only allows scaling-down the device dimension to nanometer-scale but also substantially improves the restoration repeatability and stability, promising for high-density integrated devices. For instance, ultra-small bi-stable vortex as tiny as 3 nm in diameter, can be stabilized in size-confined nanostructures[29,38], offering a possibility of developing ultra-dense memory with an areal density of 60 Tb/in$^2$. Currently, interest in 1D topological defects is seeing rapid development and highly appreciated, nonetheless exotic functionalities of these 1D defects yet remain elusive.

In this work, we demonstrate the existence of metallic conduction superfine (<3 nm) channels in two types of exotic topological defects, namely a quadrant vortex core or simply vortex core and a quadrant center domain core or simply center core, in an array of BiFeO$_3$ (BFO) nanoislands (see Fig. 1). Interestingly, these intriguing topological states can be controllably created and eliminated, leading to a remarkable electroresistance effect with an on/off ratio larger than 10$^3$ and good (thermal and fatigue-resistant) stability. The observed phenomena might open a route toward nondestructive ultrahigh density memories, based on these superfine topological objects, and offer a representative paradigm for the concept of topological electronics (or "topotronics").

## Results

In this work, BFO nanoisland arrays (~40 nm in height and 400 nm in diameter) were directly patterned from high-quality epitaxial BFO films via a nano-sphere lithography technique using the polystyrene sphere (PS) arrays as templates[10,27]. The detailed fabrication process can be found in "Methods" and Supplementary Note 1 and Supplementary Fig. 1a. The microstructure of nanoisland arrays was characterized using atomic force microscopy (AFM), X-ray diffraction (XRD), confirming that the BFO islands are highly epitaxial rhombohedral BFO phase (Supplementary Fig. 1b). The ferroelectricity of randomly selected nanoislands was examined using local piezoresponse testing, revealing butterfly-like amplitude-voltage and phase-voltage hysteresis loops (see Supplementary Fig. 1c). These characteristics indicate the high quality of these nanoislands, as further confirmed by the rather uniform and strong piezoresponse amplitude contrasts (Supplementary Fig. 2). The high quality of these nanoislands is acquired because they were obtained by low-energy Ar$^+$ beam etching and simultaneously protected by the PS template, which can substantially prevent the possible sample damage from the ion beam bombardment during the lithography process.

**Topological domain structures and conduction patterns**. To examine the domain structures, we conducted the vector piezoresponse force microscopy (PFM) measurement by recording the PFM images upon the in-plane sample rotation at 0° and 90° angles with respect to the reference direction. Consequently, the 3D domain structures from the PFM data can be reconstructed, as reported earlier[33,39]. Here, we choose two representative nanoislands for illustration: one possesses a single vortex domain state, and the other a single-center domain state.

The identification of these two types of domain structures is illustrated in Fig. 2a, b. One can derive the direction of vertical (out-of-plane) component from the vertical PFM phase image (Vert-Pha), as well as the lateral (in-plane) polarization vector maps from the lateral PFM phase images (Lat-Phase) measured for the clockwise rotation of the sample at 0° and 90° angles, which allows a determination of the local polarization components along the x axis and y axis (corresponding to the [100] axis and [010] axis of the BFO lattice), respectively, noting the fact that only eight possible [111] polarization directions are permitted by the BFO crystal symmetry. It was found that the first nanoisland contains four-quadrant head-to-tail domains (with upward vertical polarization), reflected by the uniform dark-contrast in the vertical PFM phase image), forming a lateral flux-closure vortex structure, along with four 71° neutral domain walls (NDWs) meeting at the core (see Fig. 2a). This is the typical characteristic of a vortex domain structure, as shown by the schematic vortex structure in Fig. 1b, upper panel). The second nanoisland consists of four-quadrant domains (with upward vertical polarization too). The lateral polarization components of these domains point inwards the center core, forming four head-to-head charged domain walls (CDWs) meeting at the core region, consistent with the feature of the so-called center-convergent topological state[18,33], as shown by the schematic center domain structure in Fig. 1b, lower panel). More detailed analysis of PFM amplitude/phase images for identifying of vortex state and center state are presented in Supplementary Note 2 and Supplementary Fig. 2, wherein the PFM images recorded at 0°, 90°, plus 45°, and 135° angles are included to further confirm the topological domain states.

The method described above is the conventional scheme to determine the local polarization direction, while it cannot provide more detailed information on local polarization distribution (i.e.,

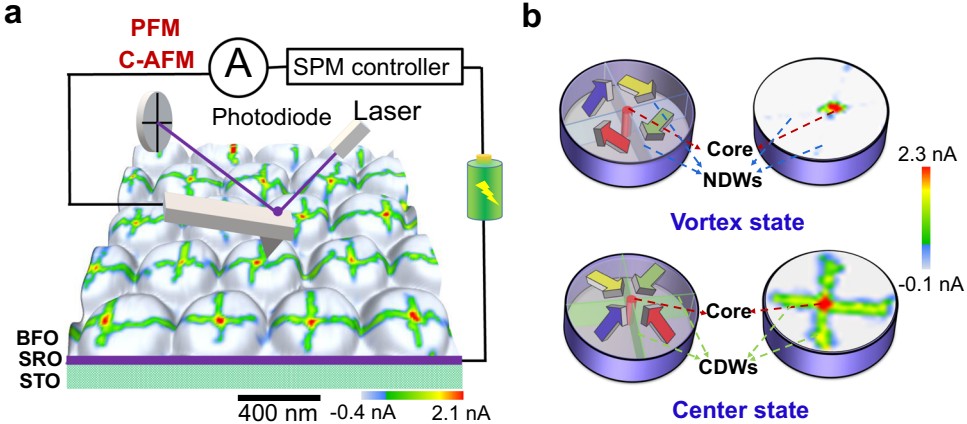

**Fig. 1 Schematic experimental setup and the typical domain structures of two types of topological states (i.e., quadrant vortex and center domain states), along with their conduction patterns. a** Schematic experimental setup for probing the C-AFM and PFM maps on an array of BFO nanoislands, wherein the 3D morphology of an array of nanoislands was superimposed with a C-AFM map for an array of center topological states. **b** Schematic domain structures of a vortex domain state and a center domain state, along with their corresponding characteristic conduction patterns (C-AFM maps). Here, Core presents a topological core (i.e., either vortex or center core), and CDWs and NDWs stand for charged domain walls and neutral domain walls (e.g., 71° head-to-tail domain walls), respectively.

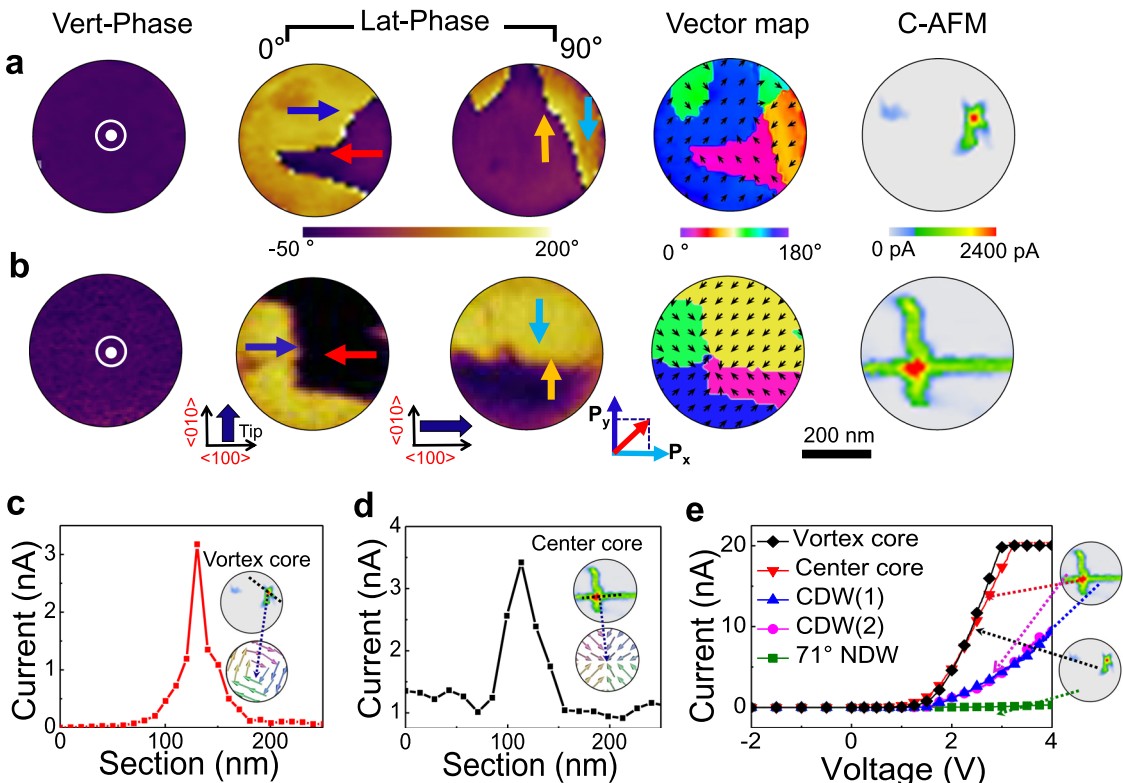

**Fig. 2 The domain structures and corresponding conductive properties for both a vortex and a center topological states confined in two nanoislands. a**, **b** PFM and C-AFM images for both a vortex state (**a**) and a center state (**b**), the micrographs from the left to the right are PFM vertical phase images illustrating the uniform upward vertical polarization components for both nanoislands, the PFM lateral phase images recorded at sample rotation of 0° and 90° to evaluate the directions of lateral polarization components respectively along $x$ axis ([100] axis) and $y$ axis ([100] axis), the lateral polarization vector direction maps derived from the PFM data, and corresponding C-AFM maps. The thick arrows aside the PFM images mark the directions of the cantilever for each PFM scan, and the fine arrows inside the images mark the directions of polarization components perpendicular to the directions of the cantilever. **c**, **d** Extracted current spatial profiles from the C-AFM maps for both the vortex (**c**) and the center (**d**) cores, extracted from **a** and **b**, respectively. The inserts in **c** and **d** illustrate the C-AFM maps and schematic local polarization configurations for the two topological cores. **e** Temperature-dependent conductive current ($I$–$V$) curves for both topological cores and domain walls.

both direction and amplitude) in sufficient accuracy. To provide a comprehensive insight into the nanoscopic polarization distribution, we conducted an angle-resolved PFM for the two topological states, following the method used in earlier literature[34,40], as shown in the Supplementary Figs. 3 and 4. In this work, the lateral piezoresponse vectors can be determined by combining the lateral PFM image data recorded at different cantilever angles for the same nanoisland.

As shown in Supplementary Fig. 3, the lateral PFM (amplitude × cos(phase)) images were first recorded by rotating the sample at nine different angles in sequence. By using trigonometric curves to fit the angular dependent piezoresponse data of each pixel (position) derived from the PFM images for the same nanoisland, one is able to determine the amplitude and phase shifts of the sinusoidal function. After obtaining the piezoresponse vectors for all the pixels in the nanoisland, one can reconstruct the lateral piezoresponse vector map, as shown in Supplementary Fig. 3d.

It was revealed that the variation of polarization (e.g., partial suppression of amplitude or rotation of vector in lateral polarization component) does occur in the central core region and adjacent to CDWs. The variation of polarization in the center core might help mitigate the effect of the uncompensated depolarization field in the core region. Similar polarization variation can also be found at regions close to charged domain walls[34,40].

From Supplementary Fig. 4, one can also see that the derived vector map for the typical vortex structure matches well with the conductive atomic force microscopy (C-AFM) map and direction map derived by the previous method. Further examination of the vectors close to the vortex core reveals that the polarization around the core shows some slight rotation and reduced amplitude of the lateral polarization component, which can greatly reduce the possible disclination strain in the core region[23]. It is also noted that the calculated polarization distribution may not be sufficiently accurate in this work, due to the possible small position shifting of raw PFM images. However, the reconstructed domain structure does reflect the local polarization distribution in a satisfactory manner.

Hereafter, for the convenience of description, the flux-closure vortex state will be abbreviated as the vortex state, and the core region is called the vortex core, while the quadrant center-convergent topological domain structure will be abbreviated as the center state, and the core region is called the center core. These domain structures are both topologically nontrivial and can be called topological states too. It is also noted that the two topological domain states can be created from the initial wedge-like domain structure by applying suitable bias voltages, and the details are described in Supplementary Fig. 5.

To illustrate the correlation between domain structure and conduction behavior, the two topological states were mapped in the C-AFM mode under a bias voltage of 2.0 V (see Fig. 2a, b). One can clearly identify the high-conduction core regions for both states, along with the relatively lower conduction paths within the cross-shaped domain walls. For the vortex state, as shown in Fig. 2a, the conduction level at the core is ~3.0 nanoamp (nA), about three orders of magnitude larger than that of the 71° NDWs whose current level is only a few picoamps (pA). Hence, the overall conductive pattern displays a highly conductive core plus four relatively faint cross-wings. In some cases, the C-AFM contrast of these cross-shaped NDWs is too faint to be visible, and thus the overall conduction pattern displays simply a bright spot at the core.

However, for the center state, the feature is somewhat different. Besides the highly conductive core region, the CDWs also exhibit rather high conductive levels (~1.0 nA). As shown in Fig. 2b, the overall conduction pattern of the center state shows four bright

wings from the conductive CDWs meeting at the even more conductive core (~3.0 nA). These unique features in the C-AFM patterns constitute the unambiguous hallmarks for the two types of states, which provide an alternative way to identify these states. For instance, the highly conductive cross-wings in the center state are consistent with earlier observations[18], nonetheless, the highest conducting core has not been previously reported.

A more precise evaluation of the conduction levels for the cores and walls of the two states can be seen from the current spatial profiles (see Fig. 2c, d), extracted from the C-AFM maps. It was found that the conduction levels for the different types of cores/walls ranking from high to low levels are: center core (~3.5 nA) ≈ vortex core (~3.3 nA) > CDWs (1.3 nA) >> 71° NDWs (~a few pA), given the identical probing bias of 2.0 V. These results can be further verified by the current–voltage (I–V) curves, obtained via placing a stationary tip on the specific core/wall region and sweeping the tip bias between −2 V and 4.0 V (as shown in Fig. 2e).

It should be mentioned that the measured I–V curves for these topological cores plus the domain walls show apparent current-rectification characteristics, whereby at negative bias range the current levels are close to noise level (see Fig. 2e). One possible reason is related to the asymmetric structure of the tip/BFO/SRO junction that leads to the asymmetric band structure and thus the current-rectification behavior. Besides, in the negative bias range, the screening electrons for stabilizing the charged cores and walls can easily run away through the tip, which may lead to the distortion (or instability) of the charged cores/walls and further reduce their conductivities. As a result, the measured current in the negative bias range must be very low.

From the above observations, it is also interested to see that the conduction level of the vortex core in this work is two orders of magnitude larger than previously reported values of artificially created vortex cores in BFO films[25], indicating a dissimilar conducting behavior. On the other hand, the conduction level of the center core is three-times larger than that of CDW, implying that the conductivity origin for the center core region is different from that of the CDW. Therefore, it is necessary to unveil the conduction mechanisms of these two types of topological cores. Moreover, the capability of deterministic tuning of these small topological objects is also a fundamental issue that deserves further attention.

**Temperature-dependent conduction behaviors**. To offer insights into the conduction mechanisms, the temperature-dependent conduction behaviors were probed (see Fig. 3). In the proceeding, a set of C-AFM images were collected in the temperature range from 25 °C to 150 °C, while their corresponding PFM images are given in the Supplementary Fig. 6. Clearly, the conduction levels of the two types of cores decrease gradually with increasing temperature, whereas the 71° NDW exhibits a monotonously increasing conduction. This trend is well-reproducible, confirmed in a number of nanoislands. It is also noted that the conductivities for both types of cores always return to the initial levels after cooling down the samples back to room temperature, indicating that these topological states are rather robust against heating.

From the C-AFM mapping, we can extract the current profiles as shown in Fig. 3c, d, and plot the current– temperature (I–T) curves accordingly (see Fig. 3e). The I–T curves for both types of cores exhibit the negative temperature coefficients, fitting to the well-known metallic conduction relation ($I \sim I_0(1 + a(T - T_0))^{-1}$), manifesting a metallic conducting behavior that is analogous to that of high-conduction CDWs[8,9]. In contrast, the NDWs exhibit a positive temperature coefficient, which is a characteristic of semiconducting behavior[9].

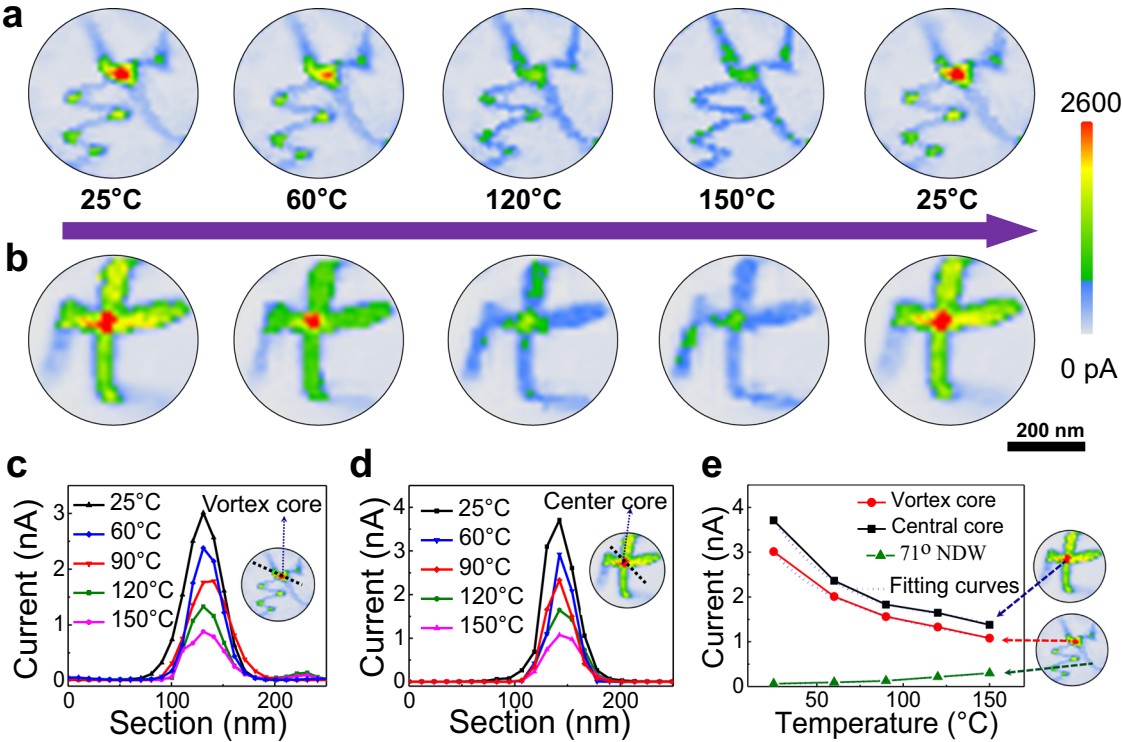

**Fig. 3 Temperature-dependent conduction behaviors for both vortex and center cores. a, b** The C-AFM maps recorded at different temperatures for two presentative nanoislands, respectively, contenting a vortex core (**a**) and a center core (**b**), along with some domain walls. The long arrow (between **a** and **b**) presents the heating and cooling consequences. **c, d** Extracted current spatial profiles from the C-AFM images (in **a, b**) as a function of temperature for the two topological cores. **e** Temperature-dependent conductive current (I–T) curves for both topological cores and a NDW wall (for comparison purpose), wherein the I–T curves for the topological cores can fit well to the metallic conductance relation ($I \sim I_0(1 + a(T - T_0))^{-1}$), while that of NDW conforms the thermal activation relation ($I \sim \exp(-\Delta E_v/kT)$).

The observed metallic conduction can be understood from the behaviors of charge carriers in the topological cores. First, the head-to-head center core contains high-density bound charges which can greatly enhance the electric potential and attract a large number of charge carriers in order to screen the bound charges. In this case, the nanoisland samples contain some extent of oxygen vacancies, as the precursor BFO films were deposited by PLD at relatively low oxygen pressure. These oxygen vacancies can serve as donor dopants with the energy level close to the conduction band of BFO. Given the charge neutrality for the whole sample, it is expected that some of the slightly trapped electrons surrounding oxygen vacancies can easily jump into the conduction band and become charge carriers, while some others remain trapped around oxygen vacancies. These activated charge carriers plus other free electrons participate in the conduction, contributing to the n-type conductivity of the BFO nanoislands prepared in this work. These carriers can also be attracted by local electric fields from the net bound charges at the head-to-head charged core/walls and accumulate at the surrounding regions to help to screen the charged core/walls. Certainly, these carriers contribute to the measured conductions of these domain walls/ cores driven by PFM tip bias.

Moreover, both the center and vortex states presented here were created by applying electric bias via the PFM tip. During the creation of center states, a large number of electrons can be injected from the conductive tip, which may help drive the formation of head-to-head charged core or CDWs on one hand, and some of these electrons can be also trapped around the charged core/walls on the other hand. After removing the electric bias/tip, the injected electrons may be partially released via thermal activation and diffuse away. Subsequently, those uncompensated bound charges (positive)

in the charged cores/walls contribute to the electric potential enhancement and attract the electrons from the domain interiors, in order to balance these charges. Consequently, energy band bending will occur, which lowers the conduction band below the Fermi levels. Eventually, the significantly enhanced conductivity and metallic conduction behaviors in the head-to-head charge core/ walls can be observed. In fact, similar conduction behaviors were reported in earlier literature addressing the charge domain walls[8–10].

To further comprehend the conduction behaviors of the two topological cores, we measured the local I–V curves at different T, as shown in Supplementary Fig. 7a, b. One can see distinctly different conduction behaviors in two different bias ranges. In the high bias range (>1.7 V), a roughly linear I–V relations can be identified, implying a metallic behavior. In the low bias range (<1.7 V), the I–V curves exhibit nonlinear behavior with a positive T-coefficient and conform to the Richardson–Schottky–Simmons emission model (Supplementary Fig. 7c–f), suggesting the thermionic emission (insulating) conducting behavior[8]. This claim can be also verified by the I–T curves measured at different bias voltages (Supplementary Fig. 7g, h), whereby the slopes of the I–T curves gradually shift from the positive value to negative value with increasing bias voltage. A metal–insulator transition for both types of cores occurs at a bias threshold of ~1.7 V.

The tip-field-driven metal–insulator transition can be interpreted from the interfacial band structure of the conduction center core (schematically illustrated in Supplementary Fig. 8). When the metallic channel contacts with the electrode, there may appear a narrow insulating gap or nonconductive domain region[9,10] close to the electrode, giving rise to the Schottky barriers between the metallic core and electrodes. Therefore, the

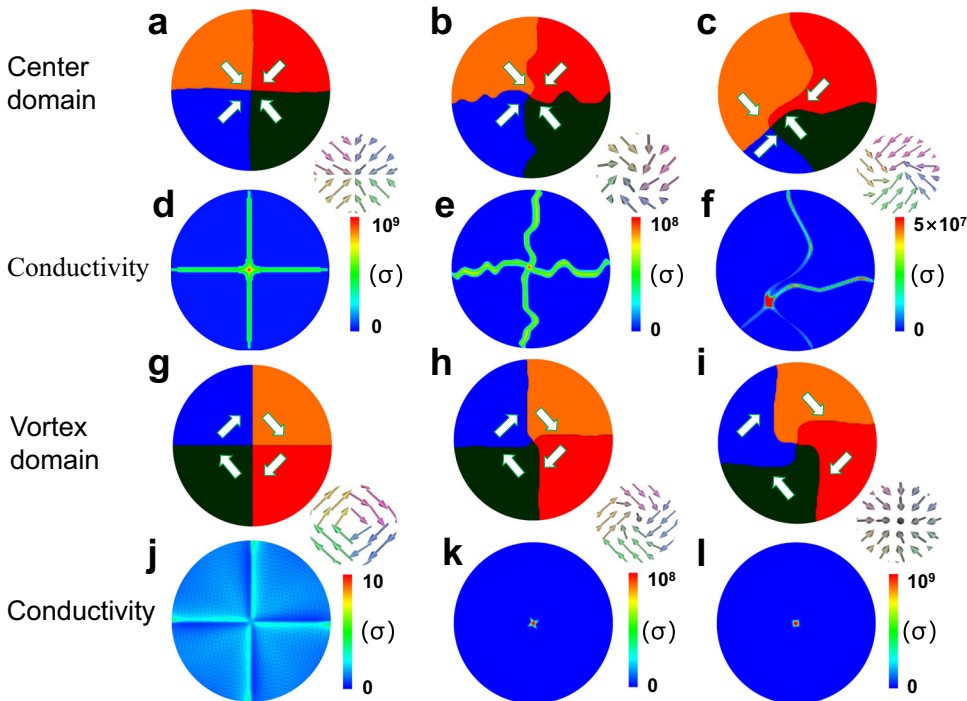

**Fig. 4 Phase-field simulation of the conduction states for both vortex and center states. a–f** Simulated domain structures and corresponding conductivity (σ) contours for the quadrant center states: perfect center domain state (**a**, **d**), the distorted center state with zigzagged charged domain walls generated via relaxing from a perfect center state (**b**, **e**), the distorted center state with curved charged domain walls formed via relaxing from an initial random polarization state (**c**, **f**). **g–l** Simulated domain structures and corresponding conductivity contours for the quadrant vortex states: perfect quadrant vortex state (**g**, **j**), distorted vortex state with a twisted core induced by applying a scanning bias of 1 V (**h**, **k**), and distorted vortex state with a severely twisted core induced by a scanning bias of 2 V (**i**, **l**). The local polarization configurations adjacent to the topological core regions were also enlarged and inset beside the individual topological domain structures.

system exhibits a thermionic emission behavior in the lower bias range.

At sufficiently large electric bias, the nonconductive domain region can be annihilated, and the large electric field significantly narrows the tunneling barrier from the insulating gap, which greatly reduces the interfacial resistance. As a result, the metallic conduction behavior of the 1D channel becomes dominant over the interfacial thermionic effect. This is consistent with the observation of a threshold in the $I–V$ curves for both types of cores, in which the nonlinear (non-Ohmic) behavior dominates below a bias voltage of 1.7 V yet linear dependence is shown at higher bias voltage (see the Supplementary Fig. 7a, b). Similar behavior was also observed in charge domain walls[8].

More interestingly, the vortex core that does not contain such conductive charged core, can also exhibit a similar metal conducting behavior. The reason for the intriguing behavior becomes an open question to be answered.

**Phase-field simulation of conduction behaviors**. To understand why these two different topological cores exhibit roughly similar conduction behaviors, we conducted the phase-field simulation. Here, the nanoislands are described as nanoscale cylinders of 70 nm in diameter (the simulation details are shown in "Methods"). Although the size of the model cylinders is smaller than the real nanoislands, nonetheless it is sufficient to study the different conduction behaviors of the two types of ultra-small cores.

Figure 4a–f shows the domain structures and conductivity contours of three center states (three columns) derived from a phase-field simulation. We first consider a perfect center domain structure without any distortion (see Fig. 4a, d), which produces a clear cross-shaped conduction pattern consisting of four CDWs

along with one small while the highly conductive core, in good agreement with the experimentally observed hallmark conductive pattern (see Fig. 1b, lower panel). After further relaxation of the domain structure, the four CDWs may become severely distorted (e.g., zigzagged), leading to an apparent nonuniformity and partial loss of the wall conductivity, while the core remains highly conductive (see Fig. 4b, e). Furthermore, given a direct relaxation from an initially random polarization state under electric fields produced by the bias near a PFM tip, in order to mimic our experimental situation, one may see bending or tilting in the CDWs to some extent, and also the nonuniform conductivity as well, noting that the high conductivity of the core region is well preserved (see Fig. 4c, f).

It is suggested that the local wall distortions (e.g., bending or tilting) can greatly redistribute the bound charges and electro-static energy, hence sizably modulating the wall conductivity[19]. The formation of the zigzagged walls rather than straight ones is driven by the release of large electrostatic energy, explaining the apparent variation and nonuniformity of the wall conductivity. In contrast, the conductivity of the center core region is rather resilient to local distortions or disturbance, manifesting the nature of topological protection. Therefore, the stable and high conduction of the center core is likely an intrinsic property of topological protection, advantageous merit for device applications.

Very differently and also unexpectedly, the simulation shows that the vortex core exhibits very low conductivity (only a rather small conductivity enhancement compared to domain interior) at zero bias field (see Fig. 4g, j), which likely contradicts our experimental result. To clarify this discrepancy, we can apply a scanning bias (1.0 V) at the core to mimic the real current reading situation, and it turns out that the conductivity of the vortex core

can be significantly enhanced via forming a twisted state (see Fig. 4h, k, and the bright dot at the center shown in Fig. 4k). This twisted state is somehow akin to a highly conductive head-to-head center core, as further supported by the observation of severely twisted core induced by 2.0 V bias (see Fig. 4i, l). Such a severely twisted core is very close to a convergent center core, while a flux-closure domain pattern is still preserved outside the core region. The twisted core was indeed observed experimentally, whereby a very large bias voltage (7.0 V) was used to stabilize it for a short while (see Supplementary Fig. 9). The aforementioned phenomena can well explain why the vortex core exhibits such conduction behavior similar to that of the center core because the measured conductivity is produced from the induced twisted vortex core stabilized during the reading process when the instant domain structure around the core was rather similar to a center core.

It was worthy of mention that the creation of such a high-conduction channel in the vortex core requires charge transfer to balance the net bound charge in the charged twisted core. During the switching process, electron carriers both from the domain interiors and injected from the conductive AFM tip can participate and thus accumulate at the core region to compensate the bound charges at the twisted core and stabilize it. Similar to the center core, the high-conduction channel in the twisted vortex core also originates from electron carriers. This process does not require the diffusion of vacancies to help to stabilize the twisted core and creating the conducting path, and thus a fast-switching speed can be expected.

To verify that the switching speed is fast, we mapped the C-AFM on the center core using different scanning speeds, as shown in (see Supplementary Fig. 10). It is clear that the high conduction in the vortex core can still be identified at a very fast scan rate of 78 Hz (the highest scan rate for our instrument), wherein the dwelling time of the tip staying at each pixel is as short as 50 μs. It implies that the switching from the vortex core to the conductive twisted state can be finished within 50 μs.

It is noted that slightly enhanced conductivity (a few pA at a reading bias of ~2.0 V) at an artificially created vortex core in BFO film was already reported previously[25], which was also interpreted by the occurrence of a metastable twisted structure that contains conductive CDW segments. To monitor the enhanced conductivity of the vortex core by C-AFM, one must pre-write the metastable twisted structure by a local bias, and the observed conductivity enhancement is rather unstable during the reading process. In contrast, the twisted vortex core in this work exhibits significantly enhanced conductivity (~3.0 nA at 2.0 V), which can be directly induced during the C-AFM scan (at scan bias of 2.0 V) without a pre-writing process. Such conductivity enhancement behavior is reproducible in different vortex cores and rather stable, likely a universal property for certain topological defects.

One can also estimate the diameters of the high-conduction center/vortex cores from the full width at half maxima of the simulated conductivity maps, and they are very small with the lateral sizes <2.5 nm (see Supplementary Fig. 11). This scale is close to the lateral size of a typical CDW[8,18], giving rise to an ultrahigh current density of ~$10^4$ A/cm². Such a superfine and highly conductive 1D metallic channel can be considered as a kind of quasi-1D electron gas (q1DEG), somewhat analogous to the quasi-two-dimensional electron gas (q2DEG) frequently observed in certain CDWs[8–10], while it is out of the scope of this work.

**Controllable creation and elimination of the high-conduction topological cores**. Certainly, it is imperative to discuss the

possible applications of these conductive topological cores in configurable devices. A fundamental issue is to achieve a controllable manipulation of these conductive topological cores from one state to the other. For this motivation, we performed extensive investigations and successfully achieved controllable and reversible creation and elimination of the vortex and center states separately for our nanoislands, via applying a suitable scanning bias with the conductive AFM tip (see Supplementary Note 8 and Supplementary Fig. 12).

The creation and deletion of the conduction channels can be clearly seen from the evolution of conduction patterns for two different topological states (corresponding PFM data can be found in Supplementary Fig. 13) in a nanoisland array. As shown in Fig. 5a, the initial pristine state usually exhibits very low conductivity (~1.0 pA) from the wedge-like domain pattern with a net downward vertical polarization (abbreviated as wedge domain state). Upon applying a scanning bias of +5.5 V on the whole array, all the nanoislands exhibit the bright cross-shaped conduction pattern which is a typical hallmark of center domains (with upward vertical polarization). After applying a negative bias of −3.5 V on four selected nanoislands (marked with red and blue circles), the selected center states switch back to the low conduction states (wedge domain state). Further application of a bias voltage of +5.5 V on two of the previously selected nanoislands creates two center states (in blue circles), and the application of a bias voltage of +3.5 V on the other two nanoislands creates two vortex states (with upward vertical polarization, in red circles) as reflected by the characteristic conduction patterns that show a single high-conduction core only. This suggests the capability of reversibly writing and deleting individual topological states as well as their conductive cores. Besides, it was also found that the vortex state can be switched to a center state by applying a bias of 5.5 V, nonetheless, this switching is not reversible (see Supplementary Fig. 12e, f).

The creation of these topological states can be attributed to the competition between the external field and internal driving forces (namely electrostatic, elastic, and polarization gradient-related energies) which are sensitive to boundary conditions and injected charges from the scanning biased AFM tip. For the case of high charge injection level (e.g., trigged by a large tip bias of 5.5 V), the injected charges can largely screen the depolarization (or electrostatic field) from the net bound charges in charged domain core/walls, and help to stabilize the center domain, while vortex state tends to be stabilized in the condition of low charge injection level (e.g. at low tip bias of 3.5 V) wherein the depolarization (electrostatic field) is dominating.

Based on the above observations, a scheme by applying a suitable bias voltage allows a programmable writing of different topological states individually as well as a programmable control of their conduction states. The topological switching process, induced by the tip bias, is schematically summarized by the triangle scheme in Fig. 5b, illustrating the capability of reversible creation/elimination of vortex and center states inside a single nanoisland, as well as the topological switching from a vortex state to a center state.

**Resistive switching properties in the topological cores**. The capability of controllable creation/elimination of these conductive states holds promises for their implementation in emerging high-density memory devices[41]. A schematic of the exemplified conceptual crossbar memory is shown in Fig. 5c, by exploiting the programmable topological states (either vortex or center state) as storage units which enable the nondestructive readout through the corresponding core conduction states. We have tested the write/read performance for such a randomly selected device, as shown in Fig. 5d, e. It is found that the resistance switching

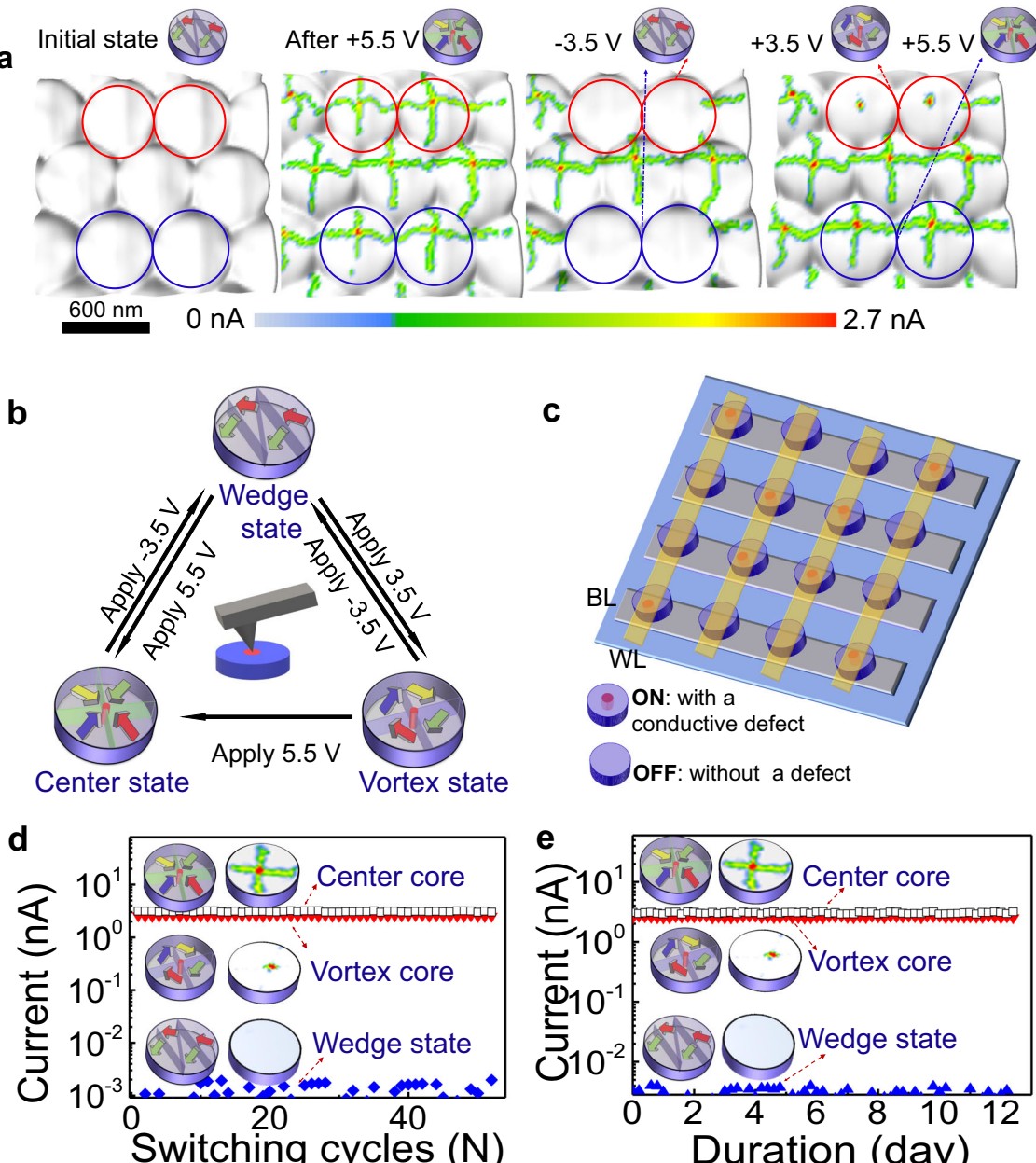

**Fig. 5 Manipulation of conductive states via controlled creation and elimination of vortex/center topological states. a** Control creation/elimination of topological states in selected nanoislands, as reflected by the hallmarks of conductive states, from left to right are: initial low conduction state with wedge domains (abbreviated as wedge states), creation of center states over the whole array by applying a scanning bias voltage of +5.5 V, elimination of center states to form low conductive wedge domain states in four selected nanoislands by applying −3.5 V, and creation of two vortex states and two center states by applying +3.5 V and +5.5 V, respectively, in the four previously selected nanoislands. **b** Schematic diagram to summarize the creation/ elimination and switching of topological states. **c** A schematic conceptual crossbar memory device using the programable metallic conduction channels in topological defects (either vortex or center cores) as data bits. Symbol WL presents the word line of the device and BL stands for bit line. **d**, **e** The repeatability (**d**) and retention (**e**) properties of resistance changes between the low and high-conduction states corresponding to writing and erasing of topological cores, which exhibit a large on/off resistance ratio over $10^3$, and can maintain stability over 50 switching cycles and retention time over 12 days ($10^7$ s), manifesting potentially high performance for the nonvolatile memory devices based on these functional topological cores.

between the low conduction wedge domain state and the high-conduction vortex/center states produces an on/off resistance ratio larger than $10^3$ which is rather stable against 50 switching cycles without apparent fatigue (see Fig. 5d). The on/off resistance states can also be maintained stable up to 12 days at room temperature (see Fig. 5e). Moreover, a retention test at 150 °C further confirms that the high-conduction levels in both cores can well preserve their stabilities against thermal perturbation for more than 7000 s (Supplementary Fig. 14), much stable compared

with the written nanodomains in ferroelectric films reported earlier[42]. The high stability of the device is mostly due to the topological protection nature of the conductive cores along with the geometric restriction of nanoislands, which enable good resilience against local disturbances or thermal fluctuations, a remarkable advantage over the 2D domain walls whose local conductivity can be sizably influenced by local disturbance[19].

Although the center core exhibits a similar conduction level to that of the vortex core, the overall conduction state of a whole

center state (including both core and CDWs) is indeed much larger compared to a vortex state, due to the contribution of much higher conductivity in the elongated CDWs. Therefore, the switching among the wedge state, vortex state, and center state also has the potential to be utilized to develop multi-level memory devices, which deserves further investigations.

On the other hand, it would be very promising if one can test the function of the device with top electrodes, nonetheless, it would become impossible to probe the underneath topological domain structures if covered with electrodes. To provide a compromised example, we placed a stationary AFM tip that serves a small top electrode on the topological cores, which mimic the structure of a solid-state device.

Then the switching property on a center core was tested. It was found that the stationary tip does enable the direct creation and deletion of the conductive channels in a central core by using electric pulses of ±6.0 V. As shown in Supplementary Fig. 15a, at a read-out bias of 2.0 V, the initial central core exhibits a high conductivity (with a high-conduction channel), and it converts to low conduction state (without a high-conduction channel) after applying an electric pulse of −6.0 V, while another electric pulse of 6.0 V can recover the high-conduction channel again. This process is repeatable for more than 20 cycles, indicating that the high conductive channel can be reversibly created and erased by electric pulses. Moreover, the resistance switching behavior also shows good retention at room temperature without apparent decline over 4 days (see Supplementary Fig. 15b). This indicates the feasibility of using such a center core in real memory devices with electrodes.

We have also conducted fatigue testing for the topological cores (see Supplementary Fig. 16). For accessing this test, we also placed the conductive tip on the topological cores and applied reversible electric pulses (voltage ± 6.0 V and pulse width 100 μs) up to $10^6$ cycles, and then collected the piezoresponse loops at different intervals, following the method employed in previous work[43,44]. The measured piezoresponse loops show that for the center core, the remanent piezoresponse signal decreases by 18% after the pulsed electric reversing for $10^6$ cycles, while the coercive field does increase by two times, reflecting that the polarization fatigue effect is nonnegligible but not so remarkable, as shown in Supplementary Fig. 16a, b. After the $10^6$ cycling test, the center core can still exhibit a well-established piezoresponse hysteresis, indicating the good fatigue-resistance, superior to the data reported earlier for ferroelectric films[43,44].

The fatigue-resistance of the center core can also be reflected by the small conductivity variation, as demonstrated by the C-AFM mapping before and after the $10^6$ fatigue cycles, as shown in Supplementary Fig. 16c. It was found that after the cycling, the conduction channel in the center core can still be reversibly created and erased, in spite of a small loss of conductivity in the low resistance state. This leads to a small reduction in the on/off resistance ratio for ~15%, further confirming the good fatigue-resistant property.

It is also noted that unlike the center core, the high-conduction channel in the vortex core does not allow reversible creation and erasure in a similar way. Once the conduction channel is broken by an electric pulse from the AFM tip, it cannot be recovered by a reversed electric pulse. The difference in the switching repeatability between the two types of cores may be attributed to the dissimilar topological protection properties between them, and additional effort is needed to improve the switching reversibility of the conduction channel in the vortex core.

Here, we would like to point out that the superfine dimension of these core conduction channels also offers an excellent possibility for scaling the device dimension down to sub-3 nm, given the sufficient capability of nondestructive current readout

from the conductive topological cores. Specifically, it was predicted that the center domain state can be stabilized at a small dimension of 16 nm[18], yet the vortex can reach a size even as small as ~3 nm[29,38]. Besides, these qusi-1D cores/topological defects are confined in nanoisland structures, which are also compatible with high-density integration processes in modern semiconductor technology.

These beneficial features create a potential pathway towards low-energy consumption, high-efficient, stable, ultra-dense, and configurable electronics devices. Specifically, the ultra-small (~3.0 nm) vortex core poses a possibility of developing nonvolatile memories with an areal density of 60 Tb per square inch[38], around four orders of magnitude higher than that of modern random-access memories. Moreover, the finding of these fascinating properties in superfine topological objects might inspire future efforts to seek other exciting unexplored properties and associated application potentials, for instance, by exploring their responses to external stimuli, e.g., strain, electric field, magnetic field, and light illumination, which might eventually enrich the field of topological electronics.

In summary, the two types of quasi-1D topological defects, namely the vortex cores and center cores, in high-quality BFO nanoislands exhibit highly conductive and metallic behaviors and behave like quasi-1D metallic conduction channels. The enhanced conductance of the center cores is an intrinsic property associated with charge carrier accumulation whereas the high conductance of the vortex cores arises from the electric field-induced twisted vortex core structure. These conductive channels can be reversibly created and deleted, producing resistance switching behaviors with an on/off ratio larger than $10^3$. This electroresistance functionality for these ultra-small 1D topological objects is stable over many cycles of switching and has a long retention time, thus it can potentially be applied to high-performance programmable topological electronic devices, e.g., ultrahigh density nonvolatile memory with nondestructive current readout.

## Methods

**Fabrication of nanodot arrays.** The fabrication procedure for the nanoisland arrays has been illustrated in the schematic flowchart in Supplementary Fig. 1, based on nano-sphere patterning on (highly) epitaxial BFO thin films. Firstly, a ~40 nm-thick epitaxial $BiFeO_3$ thin film and a ~20-nm-thick epitaxial $SrRuO_3$ bottom electrode layer were deposited on the (100)-oriented $SrTiO_3$ substrates by pulsed laser deposition (PLD). Then, the PS nanospheres pre-dispersed in a mixture of ethanol and water were transferred onto the BFO film, to form a close-packed monolayer. The sizes of the nanospheres were then shrunken by plasma etching to form a discrete ordered island array, which was followed by $Ar^+$ ion beam etching with appropriate durations. Finally, the PS template was removed by chloroformic solution and finally, the periodically ordered BFO nanoisland arrays were obtained. After the patterning, the samples were also annealed at oxygen ambiance at 400 °C.

**Microstructural characterizations.** The structures of nanoislands were characterized by X-ray diffraction (PANalytical X'Pert PRO), including $\theta$–$2\theta$ scanning and reciprocal space mapping (RSM) along with the (103) diffraction spot. The top view surface images were obtained by scanning electron microscopy (SEM, Zeise Ultra 55), and the topography images were taken by atomic force microscopy (Asylum Cypher AFM).

**PFM and C-AFM characterizations.** The ferroelectric domain structures of the nanoislands were characterized by piezoresponse force microscopy (PFM) with a scanning probe mode (Cypher, Asylum Research) using conductive PFM probes (Arrow EFM, Nanoworld). The local piezoresponse loop measurements were carried out by fixing the PFM probe on a selected nanoisland and then applying a triangle–square waveform accompanied by a small ac driven voltage from the probe. Using vector PFM mode, one can simultaneously map the vertical and lateral piezoresponse signals from the nanoisland one by one. To determine the domain structures, both the vertical and lateral PFM images were recorded at different sample rotation angles. For this, we marked the sample before the rotations, so that the same scanned area can be tracked in the different scan. The conductive current distribution maps, current–voltage (I–V) measurement were

characterized by conduct-tip atomic force microscopy (C-AFM) by using conductive probes (CDT-NCHR-10, Nanoworld).

**Phase-field simulation**. In the phase-field model, we consider both the polarization vector $P_i$ ($i = 1, 2, 3$) and the oxygen octahedral tilt vector $\theta_i$ ($i = 1, 2, 3$) as order parameters to simulate the domain patterns in BFO nanoislands[45]. The total Helmholtz free energy of BFO includes the bulk, gradient, elastic, and electrostatic free energy terms which can be written as[45–47]:

$$F = \int_V \begin{bmatrix} \alpha_{ij}\mathbf{P_i P_j} + \alpha_{ijkl}\mathbf{P_i P_j P_k P_l} + \beta_{ij}\theta_i\theta_j + t_{ijkl}\mathbf{P_i P_j}\theta_i\theta_j + \\ + \frac{1}{2}g_{ijkl}\frac{\partial \mathbf{P_i}}{\partial \mathbf{x_j}}\frac{\partial \mathbf{P_k}}{\partial \mathbf{x_l}} + \frac{1}{2}\kappa_{ijkl}\frac{\partial \theta_i}{\partial \mathbf{x_j}}\frac{\partial \theta_k}{\partial \mathbf{x_l}} + \\ \frac{1}{2}c_{ijkl}\left(\varepsilon_{ij} - \varepsilon_{ij}^0\right)\left(\varepsilon_{kl} - \varepsilon_{kl}^0\right) - \mathbf{E_i P_i} - \frac{1}{2}\varepsilon_0\kappa_b\mathbf{E_i E_j} \end{bmatrix} dV, \qquad (1)$$

where $\alpha_{ij}$, $\alpha_{ijkl}$, $\beta_{ij}$, $\beta_{ijkl}$, and $t_{ijkl}$ are the Landau polynomial coefficients. $g_{ijkl}$ and $\kappa_{ijkl}$ are the gradient energy coefficients for $P_i$ and $\theta_i$, respectively, with $x_i$ the spatial coordinate. $c_{ijkl}$ is the elastic stiffness tensor, $\varepsilon_{ij}$ is the total strain, and $\varepsilon_{ij}^0 = h_{ijkl}\,P_k\,P_l + \lambda_{ijkl}\,\theta_k\,\theta_l$ is the eigenstrain with $h_{ijkl}$ and $\lambda_{ijkl}$ the coupling coefficients. $E_i = -\partial\varphi/\partial\mathbf{x_i}$ is the electric field with $\varphi$ the electrostatic potential, $\varepsilon_0$ is the permittivity of vacuum, and $\kappa_b$ is the background relative dielectric constant. All the values of coefficients can be found in the previous literatures[46].

The temporal evolution of order parameters is simulated by the time-dependent Ginzburg–Landau equations $\partial P_i/\partial t = -L_P(\delta F/\delta P_i)$ and $\partial \theta_i/\partial t = -L_\theta(\delta F/\delta \theta_i)$ using the semi-implicit Fourier spectral method[48], where $L_P$ and $L_\theta$ are kinetic coefficients. For each time step, the elastic and electric driving forces can be calculated by solving the mechanical equilibrium equations $\sigma_{ij,j} = 0$ and the electrostatic equilibrium equation $D_{i,i} = 0$, where $\sigma_{ij}$ is the local stress and $D_i$ is the electric displacement. The spectral iterative perturbation method[49] is adopted.

The whole system grid is $256\Delta x \times 256\Delta x \times 32\Delta x$ with $\Delta x = 1.0$ nm. The system consists of three parts, i.e., $256\Delta x \times 256\Delta x \times 14\Delta x$ for the substrate, $(70\Delta x)^2\pi \times 14\Delta x$ for the BFO circular island, and the rest for air. In the air and substrate, $P_i = 0$ and $\theta_i = 0$. The elastic stiffness in the substrate is assumed the same as BFO, while the elastic stiffness in air is zero. The electric boundary conditions of the BFO islands are short-circuit for the up and bottom surfaces and open-circuit for island surroundings. The electric potential of the bottom interface $\varphi^{bot}$ is always zero, whereas the potential of the top surface $\varphi^{top}$ is uniform with certain values or nonuniform induced by a PFM tip. For the latter, it is approximated by a Lorentz distribution[25]

$$\phi^{top}(x_1, x_2) = \phi_0\left(\frac{\gamma^2}{r^2 + \gamma^2}\right), \qquad (2)$$

where $\varphi_0$ is the electric bias applied on the PFM tip, $r$ is the lateral distance from the site ($x_1, x_2$) to the position of PFM tip, and $\gamma = 15$ nm is the half-width at half-maximum of the tip.

The electrical conductivity of BFO nanoislands can be approximated from the electrostatic potential $\varphi$. If we assume that the negative charge carriers are dominant in BFO, the local conductivity can be estimated according to Boltzmann statistics as[50]

$$\sigma = N_0 e\mu \cdot \exp\left(-\frac{e\phi}{k_B T}\right), \qquad (3)$$

where $N_0$ is the background carrier density, $e$ is the charge of an electron, $\mu$ is the carrier mobility, $k_B$ is Boltzmann constant, and $T$ is the absolute temperature. To make the distributions of conductivity under different conditions comparable, we choose short-circuit boundary conditions with $\varphi^{top} = \varphi^{bot} = 0$ for certain domain structures when we calculate the conductivity.

## Data availability

The data sets that support the findings of this study are available from the corresponding authors on reasonable request.

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

## Acknowledgements

The authors would like to acknowledge the financial support from the National Key Research and Development Programs of China (Nos. 2016YFA0201002 and 2016YFA0300101), the National Natural Science Foundation of China (Nos. 11674108, 11834002, 51272078, 51721001, and 52002134), the Science and Technology Program of Guangzhou (No. 2019050001), the Natural Science Foundation of Guangdong Province (Nos. 2016A030308019 and 2019A1515110707), and the Science and Technology Planning Project of Guangdong Province (Nos. 2015B090927006 and 2019KQNCX028). Y.Z. gratefully acknowledges the financial support from China Scholarship Council (No. 201706190099). The work at Penn State is supported by the US National Science Foundation under grant number DMR-1744213.

## Author contributions

X.S.G. initiated the project and supervised the study. W.Y. and G.T. conducted the main experiments. G.T., C.C., and L.Z. fabricated the sample. W.Y. carried out the AFM, PFM, and C-AFM measurements. D.Z., Y.W., and D.C. also contributed to PFM testing. Y.Z., F.X., and L.-Q.C. conducted the phase-field simulations and analyze the data. Z.F., Z.H., J.G., M.Z., and M.Q. contributed to the data interpretation. X.S.G., L.-Q.C., and J.-M.L. conducted the data interpretation and co-wrote the article. All authors discussed the results and commented on the paper.

## Competing interests

The authors declare no competing interests.
