## [Peer Review File · Nature Communications]

REVIEWER COMMENTS

Reviewer #1 (Remarks to the Author):

The authors describe one dimensional conducting paths in BiFeO₃ nanoislands. The conducting paths appear at cores of domain vortex states and at the meeting point of four domains with the same direction of polarization (the point where 4 domains oriented head-to-head meet). They demonstrate writing and rewriting of these states which could, in principle, serve as a mechanism for high-density memories.

The work appears to be well executed and the paper is well written.

The arguments given for the actual origin of the large conductivity are hand waiving. For example, the authors write "At the head-to-head center core, there exists a large amount of uncompensated bound-charge which tends to attract a high density of electron carriers and charged point defects (e.g. vacancies)" The authors do not discuss what kind of vacancies could play a role and how they diffuse to this center point. Is usual concentration of free electrons in BiFeO₃ sufficient to compensate charge of head-to-head domains? Or does it require additional electrons that would originate from charged defects such as vacancies. Which vacancies?

On the other hand, the authors' phase field modelling does not predict high conductivity at the vortex cores. Tthe author then introduce domain twisted states and draw similarities with the center point of the head-to head domains. Considering that the vortex develops from the head-to-head domain states (if I understood correctly)and this can be controlled by the external bias, it is not clear if the charge carriers for the conductivity would be the same in vortex cores as in the 4-domains center point? If so, would vacancies diffuse to this core region to help creating conducting paths? Vacancy diffusion is a slow process that does not lend itself readily to fast switching.

So, my main issue with the paper is that there is very little discussion on the actual nature of the charge carriers in the paper although the nature of the charges is essential for the conductivity and the switching process.

Reviewer #2 (Remarks to the Author):

In their manuscript entitled "Quasi-one-dimensional metallic conduction channels in exotic ferroelectric topological defects", Yang et al. investigate the electronic conduction properties of topological defects (vortex and center states) in a BiFeO₃ nanoisland array defined by etching a continuous epitaxial layer of BiFeO₃ through polystyrene nanospheres. Vector PFM data show that applying a dc voltage while scanning the AFM tip enables a transition from the initial wedge-like domain structure to either a single vortex state (with four quadrant head-to-tail domains) or a single center domain state (with four quadrant domains pointing to the core and four head-to-head charged domain walls), depending of the amplitude of the bias voltage. Remarkably, conductive-AFM measurements reveal that both topological defects show high current (few nA) confined within the cores. Temperature-dependent measurements indicate that both defects are metallic. These results are beautifully supported by phase-field simulations. These simulations show that the center state conduction is not altered by the relaxation of the domain structure (zig-zag shaped domains), in strong contrast with the decreased conduction of the four charged domain walls, which indicates some kind of topological protection. Interestingly, the core of the vortex state is not initially conductive. The application of a dc voltage (as in experiments) leads to a twisted vortex state with high conduction at the core. Finally, the authors show that these two topological states can be created or erased repeatedly using dc voltages of

different polarities and amplitudes.

This experimental work is solid and the results are very exciting. Hence, the manuscript definitely deserves publication in Nature Communications. I just have a few corrections and comments as well as requirements to improve this manuscript and make it more appealing to the readership:

1/ At the end of the manuscript, the discussion on high-density non-volatile memories based on such topological defects (vortex and center states) implies the use of a top electrode instead of a scanning conductive AFM tip. I understand that for PFM and C-AFM measurements, BiFeO₃ nanostructures without top electrodes enable a direct connection between the domain structure and the local conductivity. To make actual solid-state devices, top electrodes are necessary and nothing proves that such kind of topological defects can be stabilized if the bias voltage is applied via a top electrode. The impact of the paper would be much higher if the authors showed the feasibility of such process.

2/ It seems that the references are misplaced and it complicates the reviewing work. The authors should double check all of them in their revised version. For example, page 4 line 4 "Skyrmions [30]". The right reference is actually [26] (Das, S. et al. Observation of room-temperature polar skyrmions. Nature 568, 368–372 (2019)). Another example, page 7 end of second paragraph: "For instance, the highly conductive cross-wings in the center state are consistent with earlier observations [18], nonetheless the highest conducting core has not been previously reported." The right reference should be [30] (Ma, J. et al. Controllable conductive readout in self assembled, topologically confined ferroelectric domain walls. Nat. Nanotechnol. 13, 947–952 (2018)).

3/ Page 12, "This is probably due to that both vortex and center states share the same winding number (+1), making it easier to convert the vortex core to a twisted state at nanometer scale." I think the authors made a mistake in this sentence as the topological charge of the vortex state is +1/2. In addition, this concept of topological charge (or number) is only valid if the polarization rotates continuously at each of the domain walls without drastic changes of the norm, is that the case? If the authors are not sure, this sentence should be removed.

Minor points (typos, lack of info, ...)

4/ It would be helpful to have the size of the nanodots (about 400 nm) in the main text.

5/ Figure 1 is very dense and some of the panels are messy. I would split it and improve the design of panels a, b and g. Especially Figure 1a is not particularly well prepared and the lateral scale is missing. (Supplementary Figure 1c is clearer).

6/ Page 6, end of first paragraph: "as shown in the schematic vortex structure in Fig. 1d" should be replaced by "as shown in the schematic center domain structure in Fig. 1d".

7/ In Supplementary Figure 2b, arrows are missing in either the lateral phase or the amplitude at 90 degrees.

8/ Caption of Figure 1: "Structures of BFO nanoslains" replace by "Structures of BFO nanoislands".

9/ Supplementary Figure 5, an "a" is missing in the "temperature" x-axis.

10/ The labels "Vortex domain" near "a" and "Center domain" near "g" are inverted in Figure 3.

11/ While mentioned in the caption the C-AFM maps are not displayed in Supplementary Figure 7.

12/ The English should be revised in the Supplementary Information.

13/ The appearance of Figure 4b could be improved.

14/ The unit of the x-axis for the retention experiments (Figure 4e) could be in days rather than seconds to make it more explicit to the reader (i.e., 11.6 days instead of 10⁷ seconds).

15/ page 23: "both the vertical and lateral PFM images were record at", replace "record" by "recorded".

Reviewer #3 (Remarks to the Author):

The authors investigated two main topological cores of a quadrant vortex and a center monopole-like domain structure in BFO island. They found that the conductance is very high in the center of both cores whereas the domain wall conductance was dependent on the degree of charging. Their findings are interesting and can be useful for future high density nonvolatile memory along with others (see e.g. Emerging Non-Volatile Memories (Eds. S. Hong, O. Auciello, D. Wouters, Springer, 2014). However, there are some significant issues that need to be addressed before judging its suitability to Nature Communications.

1. Novelty issue: Reference 11 (their own paper) contains similar figures with this manuscript. Supplementary Figure 1b and c and Figure 1a, b are almost like Figure 1a, b, c of reference 1. Figure 2 of reference 11 is similar to Supplementary Figure 3 and Figure 1e, f, g. From this similarity, it seems that the authors used the same sample and selected different islands for each paper.

2. Insufficient PFM images: First of all, they do not show any PFM amplitude images. PFM amplitude can show the degree of damage to the samples when they conduct lithography to make islands. One can immediately compare the quality of islands by comparing e.g. S. Hong et al., J. Appl. Phys. 105, 061619 (2009) [poor amplitude image, a lot of damage] vs. R. Nath et al., Appl. Phys. Lett. 96, 163101 (2010) [high amplitude image, minimal damage].

3. Monopole like core: this is thermodynamically very unstable. It is hard to believe that mere extra surface charges can screen this huge electric field in such a confined space. I would strongly encourage the authors to conduct angle-resolved PFM. Those kind of head to head domain walls were found to be zigzagged or mitigated by intermediate polarization variants [see M. Park et al., Appl. Phys. Lett. 97, 112907 (2010), M. Park et al., Appl. Phys. Lett. 99, 142909 (2011), M. Park et al., AIP Advances 3, 042114 (2013), B. Kim et al., Sci. Rep. 8: 203 (2018)].

4. Retention property: it would be ideal to image the same domain at 150 degrees Celcius to see if this new cores can really be applied to memory bits. An old example include: J. Woo et al., Appl. Phys. Lett. 80(21), 4000 – 4002 (2002).

5. Fatigue: it would be ideal to run up to at least million times or try second harmonic PFM at the same place. Examples include: E. L. Colla et al., Appl. Phys. Lett. 72(21), 2763 – 2765 (1998), N. M. Murari et al., Appl. Phys. Lett. 99, 052904 (2011).

6. There are many typos in the manuscript, which leaves a poor impression to the readers. Examples include:

A. Page 5, 17: none-destructive => non-destructive

B. Page 9: liner I-V => linear I-V

C. Page 12: an artificial created vortex => an artificially created vortex

D. Page 13: full-width of half-maximal => full-width at half maxima

E. Page 15: larger compare to => larger compared to

F. Page 22: well-epitaxial => (highly) epitaxial

G. Page 23: accompany with => accompanied by

H. Page 23: conduct probes => conductive probes

I. Page 24: consists three parts => consists of three parts, lateral distant => lateral distance

7. Figure 1g: Why didn't you try negative bias in I-V? Please add the negative bias result as well.

Reviewer #1 (Remarks to the Author):

The authors describe one dimensional conducting paths in BiFeO₃ nanoislands. The conducting paths appear at cores of domain vortex states and at the meeting point of four domains with the same direction of polarization (the point where 4 domains oriented head-to-head meet). They demonstrate writing and rewriting of these states which could, in principle, serve as a mechanism for high-density memories. The work appears to be well executed and the paper is well written.

We do appreciate very much your positive evaluation and we are very encouraged.

Q1: The arguments given for the actual origin of the large conductivity are hand waiving. For example, the authors write "At the head-to-head center core, there exists a large amount of uncompensated bound-charge which tends to attract a high density of electron carriers and charged point defects (e.g. vacancies)" The authors do not discuss what kind of vacancies could play a role and how they diffuse to this center point. Is usual concentration of free electrons in BiFeO₃ sufficient to compensate charge of head-to-head domains? Or does it require additional electrons that would originate from charged defects such as vacancies. Which vacancies?

Response:

We thank the reviewer very much for this stimulating comment. We try our best to make a detailed discussion on this issue.

First, we admit that it was not appropriate to say that the bound charges can attract charged point defects like vacancies, while in our cases the positive bound charges in the head-to-head core/wall can only attract electron carriers. Besides, the intrinsic free electrons in BFO is not sufficient to compensate the bound charges in charged cores/walls, and additional electron carriers in association with oxygen vacancies also contribute to screening of the bound charges.

In the present case, the nanoisland samples contain some extent of oxygen vacancies, as

the precursor BFO films were deposited by PLD at relatively low oxygen pressure. These oxygen vacancies can serve as donor dopant with the energy level close to the conduction band of BFO (~ 0.4 eV below the conduction band, see S. J. Clark et al, Appl. Phys. Lett. 94, 022902 (2009)). Given the charge neutrality for the whole sample, it is expected that some of the slightly trapped electrons surrounding oxygen vacancies can easily jump into the conduction band and become charge carriers, while some others remain trapped around oxygen vacancies. These activated charge carriers plus other free electrons participate in the conduction behaviors of BFO. This is the reason for the n-type conduction of the BFO nanoislands prepared in this work. These carriers can also be attracted by local electric fields from the net bound charges at the head-to-head charged core/walls and accumulate the surrounding regions to help screening the charged core/walls. Certainly, these carriers contribute to the measured conduction of these domain walls/cores driven by PFM tip bias.

It should be mentioned that the oxygen vacancies that carry positive charge cannot be attracted by the positive bound charges, although some of them may exist near these wall/core region and act as charge trappers. Therefore, only electrons in BFO (both intrinsic electron carriers and those originating from oxygen vacancies) can be attracted by the bound charge in the head-to-head charged core/walls. In this case, these charged walls/cores can be stabilized and become the conduction channels under an electric field imposed by the PFM tip for example.

Moreover, both the center and vortex domain states presented here were created by applying electric bias via the PFM tip. During the creation of center domain states, a large amount of electrons can be injected from the conductive tip, which may help driving the formation of head-to-head charged core or domain walls on one hand, and some of these electrons can be also trapped around the charged core/walls (both bound charges and possibly oxygen vacancies with positive charge) on the other hand.

After removing the electric bias/tip, the injected electrons may be partially released via thermal-activation and diffuse away. Subsequently, those uncompensated bound-charges (positive) in the charged cores/walls contribute to the electric potential enhancement and attract the electrons from the domain interiors, in order to balance these uncompensated bound charges. Consequently, energy band bending will occur, which lowers the conduction

band below the Fermi levels. Eventually, the significantly enhanced conductivity and metallic conduction behaviors in the head-to-head charge core/walls can be observed. In fact, similar conduction behaviors were reported in earlier literature addressing the charge domain walls (e.g. Sluka, T. et. al, Nat. Commun. 3, 748 (2012); Crassous, A. et. al, Nat. Nanotechnol. 10, 614–618 (2015)).

The above discussion suggests that the charges associated with the domain walls/cores include mobile (free) electrons, trapped electrons, and bound charges, and possible existing oxygen vacancies.

In the revised manuscript, we have added some more discussion regarding the contribution of charge carriers (see Page 11, Line 11-Page 12, Line 16).

Q2: On the other hand, the authors' phase field modelling does not predict high conductivity at the vortex cores. The authors then introduce domain twisted states and draw similarities with the center point of the head-to head domains. Considering that the vortex develops from the head-to-head domain states (if I understood correctly) and this can be controlled by the external bias, it is not clear if the charge carriers for the conductivity would be the same in vortex cores as in the 4-domains center point? If so, would vacancies diffuse to this core region to help creating conducting paths? Vacancy diffusion is a slow process that does not lend itself readily to fast switching.

Response:

This is a stimulating comment and we thank the reviewer for raising it.

It is true that the intrinsic conductivity of vortex core is rather low, as revealed by our simulated results. However, when we conduct the C-AFM measurement, a scanning bias of 2.0 V is usually applied onto the vortex core to read out the conduction current, which can induce the switching from a vortex core to a twisted state that is an analogue of a center core. Consequently, a vortex core can also exhibit a metallic conducting behavior analogous to the center core, as confirmed by our experimental observations and phase field simulation.

During the switching process, electron carriers both from the domain interiors and from oxygen vacancies as well as injected electrons from the conductive AFM tip can participate and thus accumulate at the core region to compensate the bound charges at the newly formed twisted core and stabilize it. Similar to the center core, the high conduction channel in the twisted vortex core also originates from electron carriers. This process does not require the diffusion of vacancies to help stabilizing the twisted core and creating the conducting path.

In a word, the creation of the conduction path mainly relies on electron movement rather than vacancy diffusion, and thus a fast switching speed can be expected.

To verify that the switching speed is fast, we mapped the C-AFM on the center core using different scanning speeds, as shown in RFig. 1 below. It is clear that the high conduction in the vortex core can still be identified at a very fast scan rate of 78 Hz (the highest scan rate for our instrument), wherein the dwelling time of the tip staying at each pixel is as short as 50 μ s. It implies that the switching from the vortex core to the conductive twisted state can be finished within 50 μ s.

In the revised manuscript, we have added RFig. 1 below as the Supplementary Fig.10 in the revised manuscript, and included some of the above discussion into the main text (see Page 16, Line 1-14).

RFig. 1. The C-AFM maps for a nanoisland with a vortex core, obtained by tip scanning at different rates with read-out bias of 2.0 V. The scan frequency and corresponding dwelling duration of the tip at each pixel are indicated below each C-AFM map.

Q3: So, my main issue with the paper is that there is very little discussion on the actual nature of the charge carriers in the paper although the nature of the charges is essential for the conductivity and the switching process.

Response:

We thank again for the reviewer's beneficial comments. The answer to this comment is presented in the response to Q1 and Q2 above. We have included some more discussion about the charge carriers and their contributions to the domain structure and domain wall/core conductivity in the revised manuscript.

Please refer to Pages 11, line 10-Page 12, line 15, and Page 16, line 1-14.

Reviewer #2 (Remarks to the Author):

In their manuscript entitled "Quasi-one-dimensional metallic conduction channels in exotic ferroelectric topological defects", Yang et al. investigate the electronic conduction properties of topological defects (vortex and center states) in a BiFeO₃ nanoisland array defined by etching a continuous epitaxial layer of BiFeO₃ through polystyrene nanospheres. Vector PFM data show that applying a dc voltage while scanning the AFM tip enables a transition from the initial wedge-like domain structure to either a single vortex state (with four quadrant head-to-tail domains) or a single center domain state (with four quadrant domains pointing to the core and four head-to-head charged domain walls), depending of the amplitude of the bias voltage. Remarkably, conductive-AFM measurements reveal that both topological defects show high current (few nA) confined within the cores. Temperature-dependent measurements indicate that both defects are metallic. These results are beautifully supported by phase-field simulations. These simulations show that the center state conduction is not altered by the relaxation of the domain structure (zig-zag shaped domains), in strong contrast with the decreased conduction of the four charged domain walls, which indicates some kind of topological protection. Interestingly, the core of the vortex state is not initially conductive. The application of a dc voltage (as in experiments) leads to a twisted vortex state with high conduction at the core. Finally, the authors show that these two topological states can be created or erased repeatedly using dc voltages of different polarities and amplitudes.

This experimental work is solid and the results are very exciting. Hence, the manuscript definitely deserves publication in Nature Communications. I just have a few corrections and comments as well as requirements to improve this manuscript and make it more appealing to the readership:

Response:

We appreciate very much the reviewer's encouragement and professional comments, and we have tried our best to revise the manuscript according to the suggestions, including performing additional experiments to clarify some issues raised by the reviewers. We hope this substantially improved version of manuscript can meet the requirements for acceptance.

Q1: At the end of the manuscript, the discussion on high-density non-volatile memories based on such topological defects (vortex and center states) implies the use of a top electrode instead of a scanning conductive AFM tip. I understand that for PFM and C-AFM measurements, BiFeO₃ nanostructures without top electrodes enable a direct connection between the domain structure and the local conductivity. To make actual solid-state devices, top electrodes are necessary and nothing proves that such kind of topological defects can be stabilized if the bias voltage is applied via a top electrode. The impact of the paper would be much higher if the authors showed the feasibility of such process.

Response:

That is definitely a wonderful comment considering the future practical applications, and we thank the reviewer very much for this comment.

We agree that it would be much promising if we could test the functionality with top electrodes. Indeed, the coverage with such a top electrode would makes it impossible to probe the domain structures underneath the electrodes. We are basically in a dilemma status from the point of view of experimental investigations. However, this issue deserves a comprehensive discussion from our experience and some compromise test.

In fact, in order to test the stability of the vortex and center states with top electrodes, we did do some experiments. For example, we placed a stationary AFM tip on the position of a topological core, which forms a specific solid-state device consisting of a stationary tip as a small top electrode, a topological defect (center or vortex core), and an SRO bottom electrode. It is noted that the top PFM tip is actually a geometry of ~ 20 nm or more in curvature radius, and it can be well touched with the top surface of the core region. It was shown from our measurements that the core/vortex domain structure can be maintained no matter how long the tip is placed onto it. We also placed the tip onto the neighboring regions of the cores and no influence on the core stability has been found. Therefore, one may say that our proto-type device should be similar to a real unit for realizing functionalities.

Furthermore, such a device structure does enable accessing to the device functionalities as well as the topological states after domain switching forward and backward by electric writing. The result revealed that the reversible erasure and creation of a conductive channel in the center core can be triggered via applying the electric pulses (± 6 V) from the stationary tip, as reflected by the significant change in conduction states of center core (see RFig. 2 below). As shown in RFig. 2a, the initial central core exhibits a rather high conductivity at a read-out bias of 2.0 V, and it converts to low conduction state (erasure of high conduction channel) upon applying an electric pulse of -6.0 V, while another pulse of +6.0 V can recover the high conduction channel again, as verified by the C-AFM maps. The reversible switching process is also repeatable for over 20 cycles.

In addition, the device also exhibits an excellent retention property, wherein both the On/Off resistance states can be maintained without an apparent decay for a few days (see RFig. 2b). The above observations demonstrate that the memory devices can operate well using the stationary AFM tip as top electrodes, which is rather close to a real solid-state device with a small top electrode, indicating the feasibility of exploiting such conductive topological defects in solid-state memory device.

It is also worth noting that, we are yet not able to reversibly switch the conduction channel in vortex core, because once the high conduction channel is erased by an electric pulse, it cannot be conveniently created by a reversed electric pulse. More efforts are needed to improve the switching reversibility in vortex core, e.g. using larger electrodes, or seeking

for the reversible switching window.

In a word, the memory device consisting of bottom electrode, center core, and stationary AFM tip as top electrode shows good memory performances, e.g. reversible writing and erasing, good retention, and nondestructive current readout. Such device structure is very close to a real solid-state device with a small electrode. As the coverage of a top electrode prevents the probing of its domain structure, such device structure provides an effective method that enable accessing to both device functionalities and domain structure. The result indicates the feasibility of exploiting the conductive topological defects in solid-state memory device.

In the revised manuscript, we have added Figure 2 below in Supplementary Fig. 15 and include some of the above discussion in the main text (see Page 19, Line 22 -Page 20, Line 15)

RFig. 2 | Device performance test with a stationary AFM tip on a center topological core.

a, Resistive switching by reversible creation and erasure of the conductive channel in center core for 20 switching cycles. The insets above **a** present the schematic device structure and C-AFM maps to illustrate the creation and erasure of the high conduction channel on the center core, triggered by applying electric pulses *via* the fixed AFM tip. **b**, Room temperature retention behavior of the device.

Q2: It seems that the references are misplaced and it complicates the reviewing work. The authors should double check all of them in their revised version. For example, page 4 line 4

“Skyrmions [30]”. The right reference is actually [26] (Das, S. et al. Observation of room-temperature polar skyrmions. Nature 568, 368–372 (2019)). Another example, page 7 end of second paragraph: “For instance, the highly conductive cross-wings in the center state are consistent with earlier observations [18], nonetheless the highest conducting core has not been previously reported.” The right reference should be [30] (Ma, J. et al. Controllable conductive readout in self assembled, topologically confined ferroelectric domain walls. Nat. Nanotechnol. 13, 947–952 (2018)).

Response:

We apologize for our carelessness and many thanks for this criticism.

In the revised manuscript, we have checked the references one by one and corrected those mistakes.

Q3: Page 12, “This is probably due to that both vortex and center states share the same winding number (+1), making it easier to convert the vortex core to a twisted state at nanometer scale.” I think the authors made a mistake in this sentence as the topological charge of the vortex state is $+1/2$. In addition, this concept of topological charge (or number) is only valid if the polarization rotates continuously at each of the domain walls without drastic changes of the norm, is that the case? If the authors are not sure, this sentence should be removed.

Response:

Many thanks for this comment and indeed it is improper to make such argument without solid evidence.

We have removed the sentence “This is probably due to that both vortex and center states share the same winding number (+1), making it easier to convert the vortex core to a twisted state at nanometer scale” from the main text in the revised version.

Q4: Minor points (typos, lack of info, ...)

- a/ It would be helpful to have the size of the nanodots (about 400 nm) in the main text.
- b/ Figure 1 is very dense and some of the panels are messy. I would split it and improve the design of panels a, b and g. Especially Figure 1a is not particularly well prepared and the lateral scale is missing. (Supplementary Figure 1c is clearer).
- c/ Page 6, end of first paragraph: “as shown in the schematic vortex structure in Fig. 1d” should be replaced by “as shown in the schematic center domain structure in Fig. 1d”.
- d/ In Supplementary Figure 2b, arrows are missing in either the lateral phase or the amplitude at 90 degrees.
- e/ Caption of Figure 1: “Structures of BFO nanoslains” replace by “Structures of BFO nanoislands”
- f/ Supplementary Figure 5, an “a” is missing in the “temperature” x-axis.
- g/ The labels “Vortex domain” near “a” and “Center domain” near “g” are inverted in Figure 3
- h/ While mentioned in the caption the C-AFM maps are not displayed in Supplementary Figure 7
- i/ The English should be revised in the Supplementary Information.
- j/ The appearance of Figure 4b could be improved.
- k/ The unit of the x-axis for the retention experiments (Figure 4e) could be in days rather than seconds to make it more explicit to the reader (i.e., 11.6 days instead of 10^7 seconds).
- l/ page 23: “both the vertical and lateral PFM images were record at”, replace “record” by “recorded”.

Response:

Many thanks for such careful checking of these typos and improper text which should have been already corrected from our side. We apologize for them.

We have corrected the typos and made some modification in figures. Particularly, we have made some major changes to Fig. 1 and separated it into two figures to avoid the messy panels, and Figure 4b was also slightly modified.

The whole manuscript including the Supplementary is carefully checked for English writing, and hopefully the quality of manuscript and figures can be much improved.

Reviewer #3 (Remarks to the Author):

The authors investigated two main topological cores of a quadrant vortex and a center monopole-like domain structure in BFO island. They found that the conductance is very high in the center of both cores whereas the domain wall conductance was dependent on the degree of charging. Their findings are interesting and can be useful for future high density nonvolatile memory along with others (see e.g. Emerging Non-Volatile Memories (Eds. S. Hong, O. Auciello, D. Wouters, Springer, 2014). However, there are some significant issues that need to be addressed before judging its suitability to Nature Communications.

Response:

We really appreciate the reviewer's encouraging assessment and professional comments. We have included some new experimental data as suggested, made substantial revision, and provide point-to-point responses to the queries. Besides, we have quoted some of the recommended literature in the reference list. Hopefully the quality of our manuscript can be improved to a satisfactory level for Nature Communications.

Q1: Novelty issue: Reference 11 (their own paper) contains similar figures with this manuscript. Supplementary Figure 1b and c and Figure 1a, b are almost like Figure 1a, b, c of reference 1. Figure 2 of reference 11 is similar to Supplementary Figure 3 and Figure 1e, f, g. From this similarity, it seems that the authors used the same sample and selected different islands for each paper.

Response:

Many thanks for this comment which is critical since readers would have similar questions if no clear explanation is made.

We agree that the figures in this work show some similarities with those appeared in reference 11 (Ref. 10 in the revised manuscript). While the samples used in the present work and Ref. 11 were fabricated using similar techniques, *i.e.* template assist nano-patterning from well epitaxial BFO films deposited by PLD, it is not strange to find some similarities, e.g. X-ray diffraction pattern and AFM morphologies.

However, it should be mentioned that the samples used in the two works are not the same. In fact, checking shows that the domain structures are rather different, and specifically the topological domain states can be controllably created using the sample in this work, while the samples reported in Ref. 11 did not work so easily.

As the domain switching behaviors are very sensitive to the quality of films (e.g. defect states, strain, and so on), our experience shows that a relatively small variation in the fabrication parameters may lead to sizable difference of the samples' properties.

We also agree that the manner of data presentation in Fig. 1e, f, g in this work is somehow similar to that of Fig. 2 in Ref. 11. While the data in Fig. 1 of this work focus on topological cores and the results in Fig. 2 of Ref.11 emphasize on the conductivities of domain walls, one can see the apparent difference in the current levels between these two figures.

Besides, the Supplementary Fig. 3 in the present work is different from Fig. 2 of Ref. 11, in that the former mainly shows the different exotic topological states and the latter is related with normal domain structures.

In the revised manuscript, we made some modifications to the figures in this work to weaken the similarities.

Q2: Insufficient PFM images: First of all, they do not show any PFM amplitude images. PFM amplitude can show the degree of damage to the samples when they conduct lithography to make islands. One can immediately compare the quality of islands by comparing e.g. S. Hong et al., J. Appl. Phys. 105, 061619 (2009) [poor amplitude image, a lot of damage] vs. R. Nath et al., Appl. Phys. Lett. 96, 163101 (2010) [high amplitude image, minimal damage].

Response:

Many thanks for this suggestive comment.

In fact, we have all these amplitude images and they show high quality confirming the high quality of the nanoislands prepared in our work. The PFM amplitude image for a representative nanoisland can be found in RFig. 3 below (also in the Supplementary Fig. 2 in manuscript). It was found that the amplitude images show rather uniform and strong piezoresponse signals over the whole nanoisland, except the dark-contrast lines marking the domain walls. It is thus indicated that the quality of the nanoisland in the present work is much better than those from BFO nanoislands directly etched by focused ion beam (FIB) technique (S. Hong et al., *J. Appl. Phys.* 105, 061619 (2009)), and are comparable with those BFO islands subjected to an annealing process (refer to. R. Nath et al., *Appl. Phys. Lett.* 96, 163101 (2010)).

The high-quality of the sample in the present work can be understood by the fact that the nanoislands were well protected by the PS nano-sphere template during the etching process, which can greatly avoid the possible damage from the lithography process. Moreover, unlike the focused ion beam patterning technique that employs high energy Ga^+ beam (with acceleration voltage of 30 kV, see S. Hong et al., *J. Appl. Phys.* 105, 061619 (2009)), here much lower energy (with acceleration voltage of 300 V) Ar^+ ion beam was used in the present work, which can further reduce the sample damages from the ion bombarding process. The good sample quality can be verified by the well-established piezoresponse hysteresis loops observed in the patterned nanoislands (refer to the Supplementary Figure 1).

In the revised manuscript, we have added some discussions about sample quality in the revised manuscript (Page 5, Line 15-20).

RFig. 3 | The domain structure reconstruction for a vortex domain and a center domain by conventional method. a,b, The lateral polarization vector maps and their corresponding C-AFM maps, the vertical PFM phase (V-Pha) and amplitude (V-Amp) images, and the lateral PFM phase (L-Pha) and amplitude (L-Amp) images captured by sample rotation for various angles (0° , 90° , 45° , 135°), for the vortex state (a) and center state (b). The big blue arrows outside images mark the cantilever direction, and the small arrows inside images indicate the lateral local orientations of polarization components perpendicular to the cantilever. The double-head arrows illustrate the non-defined directions of local polarization components parallel to the cantilever, given that the marked regions show dark contrast in the lateral amplitude image.

Q3: Monopole like core: this is thermodynamically very unstable. It is hard to believe that mere extra surface charges can screen this huge electric field in such a confined space. I would strongly encourage the authors to conduct angle-resolved PFM. Those kind of head to head domain walls were found to be zigzagged or mitigated by intermediate polarization variants [see M. Park et al., Appl. Phys. Lett. 97, 112907 (2010), M. Park et al., Appl. Phys. Lett. 99, 142909 (2011), M. Park et al., AIP Advances 3, 042114 (2013), B. Kim et al., Sci.

Rep. 8: 203 (2018)].

Response:

Many thanks for this comment and it is stimulating.

We do agree that a non-screened monopole like core is rather unstable, while the large depolarization in the core region can be screened by electron charge carriers inside nanoislands as well as injected electrons from conductive AFM tip, which can accumulate at the core region to help compensating the bound charges, as mentioned previously.

As the referee recommended, we conducted the angle-resolved PFM to map the microscopic polarization distribution for a center state, following the method developed in earlier literature (e.g. M. Park et al., Appl. Phys. Lett. 97, 112907 (2010), Kim, K. E. et al. Nat. Commun. 9, 403 (2018)), as shown in RFig. 4 below. In this work, the lateral piezoresponse vectors can be determined *via* combining the lateral PFM (Lat-PFM) images for the same nanoisland scanned at different cantilever-orientation angles. As shown in RFig. 4a, the Lat-PFM (amplitude*cos(phase)) images were recorded by rotating the sample at 9 different angles. By using a trigonometric fitting method on the angular-dependent piezoresponse data of each pixel (position) from the PFM images, one is able to determine the amplitude and phase shift of the sinusoidal function.

In this way, the local distribution of lateral piezoresponse vectors (with amplitude and phase) for the whole nanoisland can be obtained. For example, the trigonometric curve fitting for the four selected positions was conducted using the local piezoresponse data extracted from the Lat-PFM images at 9 different angles (see RFig. 4c), and consequently the phase shift and amplitude can be derived. After obtaining the piezoresponse vector for all the pixels in the nanoisland, one can draw a lateral piezoresponse vector map (shown in RFig. 4d), which exhibits a typical center domain structure, agreeing with the vector direction map derived from a conventional method and matching well with the C-AFM maps.

It was revealed that some variation of polarization (e.g. partial suppress of amplitude or rotation of vector in lateral component of polarization) do occur in the center core region and adjacent to CDWs, in agreement with what have reported in CDWs by Park et. al and Kim et. al. The variation of polarization in the center core might help mitigate the effect of

uncompensated depolarization field in the core region. It is also noted that the calculated polarization distribution may not be sufficiently accurate in this work, due to the possible small position shifting of raw PFM images. However, the reconstructed domain structure does reflect in a satisfactory manner the local polarization distribution.

In the revised manuscript, we have added the angle-resolved PFM analysis for both the vortex and center topological states in the Supplementary materials (Fig. 3 and Fig. 4), and also have included some discussion in the main text (see Page 7, Lines 3-Page, 8, line 9).

RFig. 4 | The angle-resolved lateral PFM images used to reconstruct the polarization vector map for a selected center domain state. a, The lateral PFM images (amplitude*cos(phase)) scanned at 9 different rotation angles for the same nanoisland. **b,** A schematic diagram showing the cantilever orientation for each individual scan, a simplified lateral direction map (without amplitude) of the nanoisland derived from the conventional method, and the corresponding C-AFM map. **c,** Examples of the trigonometric fitting curves

for the four selected positions using the local piezoresponse data extracted from the Lat-PFM images scanned at 9 different angles, and these data are used to determine both the amplitude and orientation of the lateral local piezoresponse vectors. **d**, The reconstructed lateral polarization vector maps for the center state in color contrast and arrow configuration, along with a magnified vector map for the center core region.

Q4: Retention property: it would be ideal to image the same domain at 150 degrees Celcius to see if this new cores can really be applied to memory bits. An old example include: J. Woo et al., Appl. Phys. Lett. 80(21), 4000 – 4002 (2002).

Response:

Many thanks for this suggestion.

Following this suggestion, we have conducted a retention test for both the center and vortex states at 150 °C for over 7200 s (120 min), and the conduction current data in the cores against the duration time was recorded (see RFig. 5 below). It was revealed that the two types of cores are rather stable against the high temperature annealing without apparent conduction decaying (RFig. 5a). This was also verified by the C-AFM maps for the two states before and after the retention test (RFig. 5b). The retention property is much better than those written nanodomains in a Pb(Zr,Ti)O₃ thin film (J. Woo et al., Appl. Phys. Lett. 80(21), 4000 – 4002 (2002)). This enhanced retention stability of these topological cores can be attributed to the topological protection and geometric restriction effects.

In the revised manuscript, we have included some more discussion in the main text (Page 19, Lines 8-11), and added RFig. 5 here into the Supplementary Materials as Fig. 14.

RFig. 5 | The retention behaviors for the vortex and central cores at 150 °C. **a**, The time dependent conduction current at 150 °C for over 7200 s (120 min) for the two types of topological cores. **b**, The C-AFM maps for the two cores before and after the retention test.

Q5: Fatigue: it would be ideal to run up to at least million times or try second harmonic PFM at the same place. Examples include: E. L. Colla et al., Appl. Phys. Lett. 72(21), 2763 – 2765 (1998), N. M. Murari et al., Appl. Phys. Lett. 99, 052904 (2011).

Response:

Many thanks for this comment and suggestion.

We have conducted the suggested fatigue testing for the topological cores (see RFig. 6 below). For accessing this test, we placed the conductive tip on the nanoisland and applied reversible electric pulses (voltage $\pm 6.0V$ and pulse width 100 μs) up to 10^6 cycles, and then collected the piezoresponse loops at different intervals, following the method employed in previous works (e.g. E. L. Colla et al., Appl. Phys. Lett. 72(21), 2763 – 2765 (1998), N. M. Murari et al., Appl. Phys. Lett. 99, 052904 (2011)). For simplicity, we only examined the fatigue behaviors of the topological cores.

The measured piezoresponse loops show that for the center core, the remanent piezoresponse signal decreases by 18% after the pulsed electric reversing for 10^6 cycles, while the coercive field does increase by two times, reflecting that the polarization fatigue effect is non-negligible but not so remarkable, as shown in RFig. 6a and 6b. After the 10^6 cycling test,

the center core can still exhibit a well-established piezoresponse hysteresis, indicating the good fatigue-resistance, much better than the data reported earlier for ferroelectric films (e.g. E. L. Colla et al., Appl. Phys. Lett. 72(21), 2763 – 2765 (1998), N. M. Murari et al., Appl. Phys. Lett. 99, 052904 (2011)).

The fatigue-resistance of the center core can also be reflected by the small conductivity variation, as demonstrated by the C-AFM mapping before and after the 10^6 fatigue cycles, as shown in RFig. 6c. It was found that after the cycling, the conduction channels in the center core can still be reversibly created and erased, in spite of a small loss of conductivity in the low resistance state. This leads to a small reduction in the On/Off resistance ratio for $\sim 15\%$, further confirming the good fatigue-resistant property.

It is noted that different from the center core, the vortex core does not allow creation and erasure of high conduction channel in a similar way. Once the conduction channel is damaged by an electric pulse from the AFM tip, it cannot be recovered by a reversed electric pulse. This difference in the switching repeatability between the two types of cores may be attributed to the dissimilar topological protection properties between them, and additional effort is needed to improve the switching reversibility of the conduction channel for the vortex core (e.g. using larger electrode or finding a suitable switching window).

In the revised manuscript, we have included some more discussion done above (Page 20, Line 16 - Page 21, Line 18), and added RFig. 6 below into the Supplementary as Figure 16.

RFig. 6 | Fatigue-resistance behaviors for a center core. a, Piezoresponse hysteresis loops measured on the center core after pulsed electric reversing for different cycles up to 10^6 cycles. **b**, Gradual decrease in the piezoresponse amplitude against the fatigue cycling. **c**, The C-AFM maps of the center domain states before and after the 10^6 fatigue test cycles. LRS

stands for the low resistance state (with a high conduction channel) and HRS for the high resistance state (erasure of the conductive channel) in the center core.

Q6: There are many typos in the manuscript, which leaves a poor impression to the readers.

Examples include:

a/ Page 5, 17: none-destructive => non-destructive

b/ Page 9: liner I-V => linear I-V

c/ Page 12: an artificial created vortex => an artificially created vortex

d/ Page 13: full-width of half-maximal => full-width at half maxima

e/ Page 15: larger compare to => larger compared to

f/ Page 22: well-epitaxial => (highly) epitaxial

g/ Page 23: accompany with => accompanied by

h/ Page 23: conduct probes => conductive probes

i/ Page 24: consists three parts => consists of three parts, lateral distant => lateral distance

Response:

Many thanks for the reviewer's careful checking, and we have corrected the typos accordingly.

Q7: Figure 1g: Why didn't you try negative bias in I-V? Please add the negative bias result as well.

Response:

Following this comment, we have added the I - V data in the negative bias range into Fig. 1g (also refer to RFig. 7 below). It was found that the I - V curves for the two types of cores plus the charged domain walls exhibit the apparent current rectification characteristics. In the negative bias range, the current levels are rather low (close to the noise level). One possible reason is the asymmetric structure of Pt/BFO/Au junction. The as-generated asymmetric band

profile (or built-in voltage) results in the current rectification behaviors.

Besides, for the negative bias probing, the screening electrons (e.g. injected charges) that stabilize the head-to-head charged cores and walls can be easily removed away through the PFM tip. This effect would distort or even destroy the cores/domain walls, thus removing the conduction channels. As a result, the measured current I in the negative bias range must be very low.

In this revised version, we have added the relevant discussion into the manuscript (see Page 10, Lines 7-15).

RFig. 7 | The measured $I - V$ curves for different topological cores and domain walls, including the curves measured in the negative bias range.

REVIEWERS' COMMENTS

Reviewer #1 (Remarks to the Author):

The authors have revised the paper by addressing all the issues I have raised. I can now recommend the paper for publication.

Reviewer #2 (Remarks to the Author):

I have carefully read the rebuttal letter and the revised manuscript of Yang et al. entitled "Quasi-one-dimensional metallic conduction channels in exotic ferroelectric topological defects". I am satisfied by the way the authors addressed the different points raised by the other two Referees and myself. In addition, the manuscript now contains important additional data in the supplementary material, such as the vector PFM amplitude and phase, angle-resolved PFM, retention experiments at 150°C and fatigue experiments up to 10^6 cycles. Therefore, I recommend the publication of this work in Nature Communications.

I just have one minor point that requires further clarification: the authors did not mention if the AFM scanning direction (along [100], [110], [010], ...) is critical during the writing process of the vortex and center states (as it is to write in-plane components of the polarization in BiFeO₃ thin films with the trailing field). Is it something that the authors looked at?

Typo line 483: "charge caries" -> "charge carriers"

Reviewer #1 (Remarks to the Author):

The authors have revised the paper by addressing all the issues I have raised. I can now recommend the paper for publication.

Response: We thank very much the reviewer for his very positive and encouraging recommendation.

Reviewer #2 (Remarks to the Author):

I have carefully read the rebuttal letter and the revised manuscript of Yang et al. entitled “Quasi-one-dimensional metallic conduction channels in exotic ferroelectric topological defects”. I am satisfied by the way the authors addressed the different points raised by the other two Referees and myself. In addition, the manuscript now contains important additional data in the supplementary material, such as the vector PFM amplitude and phase, angle-resolved PFM, retention experiments at 150°C and fatigue experiments up to 10^6 cycles. Therefore, I recommend the publication of this work in Nature Communications.

We do appreciate very much the reviewer’s recommendation.

Q1: The authors did not mention if the AFM scanning direction (along [100], [110], [010], ...) is critical during the writing process of the vortex and center states (as it is to write in-plane

components of the polarization in BiFeO₃ thin films with the trailing field). Is it something that the authors looked at?

Response: Yes, we agree that the AFM scanning direction is very important for tuning the domain structure in BFO film, which generate in-plane trailing field of different directions. However, in the present case of nanoislands, the creation of vortex and center states does not apparently depend on the scanning directions. As shown in RFig. 1 below, very similar center domain states can be created by separate scanning sequences along three different scanning directions (along [100], [110], and [010] directions). The creation of vortex state also follows a similar trend, although the stabilization window is narrower.

The reason for the above observations can be interpreted by the competition between the effect of external field and internal driving forces (consisting of electrostatic, elastic, and polarization gradient-related energies) which are very sensitive to boundary conditions of nanoisland and density of injected charges from the scanning AFM tip. It is true that in BFO film, the effect of trailing field plays some roles, which can effectively create different domain structures. Unlike the case of film, in nanoscale BFO islands, the internal driving forces tend to outweigh the trailing field from the tip, leading to the formation of vortex and center states depending on different charge injection levels but much less depending on the tip trailing field. For the case of high charge injection level (e.g. triggered by large tip bias of 5.5 V), the injected charges can largely screen the depolarization (or electrostatic field) from the net bound charges in charged domain core/walls, and help to stabilize the center domain, while vortex state tends to be stabilized in condition of low charge injection level (with low tip bias of 3.5 V) wherein the depolarization (electrostatic field) is dominating. In a word, both the size effect and level of trapped charges play more important roles in the formation of these topological states, than the trailing field. As a result, tip scanning direction cannot apparently affect the formation of these topological states in the present work.

In the revised manuscript, we have added some of the above discussions (Page 18, Line 15-22).

RFig. 2. **Creation of center domain states by tip scanning along different scanning directions.** The three panels from left to right present the lateral PFM phase images (recorded at both 0° and 90°) written at three scanning directions along $[010]$, $[110]$, and $[100]$ directions, respectively. One can see that in the case of nanoislands, the direction of tip movement does not apparently affect the final product of the created central domains.

Q2: Typo line 483: “charge caries” -> “charge carriers”

Response: We thank the reviewer again for carefully checking the manuscript. We apologize for the typo, and have corrected it.